# ToolRM: Outcome Reward Models for Tool-Calling Large Language Models

## Abstract

As large language models (LLMs) increasingly interact with external tools, reward modeling for tool use has become a critical yet underexplored area. Existing reward models, trained primarily on natural language outputs, struggle to evaluate tool-based reasoning and execution. To quantify this gap, we introduce FC-RewardBench, the first benchmark designed to systematically evaluate reward models in tool-calling scenarios. Our analysis shows that current reward models often miss key signals of effective tool use, highlighting the need for domain-specific modeling. To address this, we propose a training framework for outcome reward models using data synthesized from permissively licensed, open-weight LLMs. We train models ranging from 1.7B to 14B parameters and evaluate them across seven out-of-domain benchmarks. These reward models consistently outperform general-purpose baselines, yielding up to a 25% average improvement in downstream task performance, enhancing robustness to input noise, and enabling data-efficient fine-tuning through reward-guided filtering.

## 1 Introduction

Large language models (LLMs) such as GPT-4 (Achiam et al., 2023), Claude, and Gemini (Reid et al., 2024) have rapidly advanced the field of artificial intelligence (AI), achieving strong performance across a wide range of tasks, including complex question answering, code generation, and multi-step reasoning (Li et al., 2025b). As these models are increasingly deployed in real-world systems, the need for them to interact with external tools has become critical. Tool calling enables LLMs to invoke external functions such as APIs, databases, calculators, and search engines (Prabhakar et al., 2025b; Zhang et al., 2024; Abdelaziz et al., 2024; Liu et al., 2024b; Lin et al., 2024), shifting their role from standalone text generators to orchestrators of complex workflows. This capability underpins their application in autonomous agents, virtual assistants, and multimodal systems.

Training these LLMs effectively requires reward models, which are integrated into the learning pipeline through reinforcement learning (RL), preference optimization (Wang et al., 2023), and rejection sampling fine-tuning (Touvron et al., 2023; Team, 2024). Reward models provide learned signals that estimate output quality, enabling scalable evaluation without requiring human judgment on every example. Broadly, they fall into two categories: process reward models (PRMs) (Lightman et al., 2023), which score intermediate reasoning steps, and outcome reward models (ORMs) (Cobbe et al., 2021), which evaluate only the final answer. PRMs offer finer control over reasoning but demand costly, fine-grained annotations. ORMs, in contrast, are easier to train and can achieve comparable performance gains (Uesato et al., 2022).

Despite their successes, current reward models are designed primarily for natural language outputs (Zhong et al., 2025). Reward modeling for tool calling remains an underexplored area, with two notable gaps: (a) no dedicated benchmark exists for evaluating reward models in the function-calling domain[1], and (b) existing reward models fail to capture the nuances of tool-based reasoning and execution. In order to address these gaps, we first introduce FC-RewardBench – a comprehensive benchmark specifically designed to evaluate reward models on tool-calling tasks. Derived from the Berkeley Function Calling Leaderboard (BFCL) Version 3 (Patil et al., 2025), the dataset contains

---

[1]Tool-use, tool-calling, and function-calling are used interchangeably throughout the paper

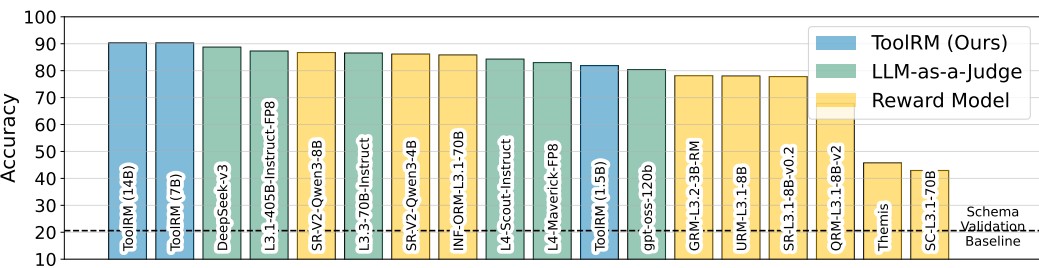

Figure 1: Performance of ToolRM, top reward models from RewardBench, Tool-augmented RM (Themis), and leading LLMs-as-judges on FC-RewardBench. *Note:* Model names are abbreviated for conciseness (e.g., L3.1-xx, SR-xx, and SC-xx correspond to Llama-3.1-xx, SkyWorks-Reward-xx, and SkyWorks-Critics-xx, respectively). Full model names are provided in Appendix A.4.

1500 user inputs paired with correct and incorrect function calls. We benchmark several state-of-the-art general-purpose reward models on FC-RewardBench, and our analysis (Figure 1) shows that these models often fail to capture key aspects of successful tool use, hence failing to capture the nuances of tool-based reasoning and execution. To this end, next, we introduce ToolRM, a collection of specialized ORMs for tool calling. Trained on preference data synthesized from a diverse set of open-source function-calling models, ToolRM outperforms much larger reward models and LLMs-as-Judges on FC-RewardBench. In downstream applications, ToolRM demonstrates up to 25% average improvement across multiple benchmarks in a Best-of-$n$ setting. These models also enable efficient data filtering, yielding better fine-tuned models with less data.

In summary, our contributions are:

- We introduce FC-RewardBench; the first benchmark to evaluate reward models on a tool calling setting with strong correlation to downstream task performance.
- We propose a framework for training an ORM for tool-calling use-case using data generated from permissively-licensed, open-weight LLMs.
- We train multiple reward models (ToolRM) varying in size from 1.7B to 14B in parameters, and extensively evaluate our proposed models on seven out-of-domain benchmarks.
- We evaluated ToolRM against greedy decoding, majority voting, and rule-based schema validation in a Best-of-$n$ ($n = 32$) setting, across diverse tool-calling models of varying sizes. They yield up to a 25% average benchmark improvement and enable efficient data filtering for better finetuned models with less data.

## 2 RELATED WORK

### 2.1 TOOL CALLING

Tool calling has extended LLMs beyond static knowledge to tasks requiring external retrieval (Schick et al., 2023), reasoning (He-Yueya et al., 2023), orchestration (Jain et al., 2024), and code execution (Gao et al., 2023). Early prompting-based approaches such as ReAct (Yao et al., 2023) inspired refinements for efficiency (Xu et al., 2023), performance (Shinn et al., 2023; Yang et al., 2023), or balanced trade-offs (Crouse et al., 2023). Recent models now provide built-in tool use (Reid et al., 2024; CodeGemma Team et al., 2024; CohereForAI, 2024; AI@Meta, 2024; Jiang et al., 2023) or are fine-tuned for this capability (Qin et al., 2023; Tang et al., 2023; Patil et al., 2023; Abdelaziz et al., 2024). To assess and enhance these capabilities, benchmarks (Guo et al., 2024; Patil et al., 2023), curated datasets (Liu et al., 2024b; Qian et al., 2025b), and autonomous tool construction methods (Qian et al., 2023b;a) have been proposed.

### 2.2 RL FOR TOOL-USE ALIGNMENT

Reinforcement Learning has become a powerful approach for aligning LLMs with effective tool use. Search-R1 (Jin et al., 2025) trains LLMs to iteratively refine search queries, showing RL feedback

balances exploration and retrieval precision. ToRL (Li et al., 2025a) enables models to discover tool-use strategies autonomously, with rewards driving emergent behaviors like strategic invocation and adaptive reasoning mode switching. ReTool (Feng et al., 2025) interleaves code execution with natural language reasoning, using outcome feedback to guide tool invocation, improving mathematical problem solving. Several works focus on reward design: ToolRL (Qian et al., 2025a) studies how reward type, granularity, and temporal dynamics affect alignment; StepTool (Yu et al., 2024) uses step-level reward shaping and policy-gradient optimization for multi-step tasks; CodeTool (Lu et al., 2025) combines RL with step-level supervision to encourage reasoning about intermediate states; SWE-RL (Wei et al., 2025) leverages software evolution data to optimize reasoning over action sequences, capturing temporal dependencies; and iTool (Zeng et al., 2025) mitigates performance decay from synthetic data via iterative reinforced fine-tuning with Monte Carlo Tree Search, enhancing robustness. Together, these works show RL's effectiveness in aligning LLMs for general-purpose tool use, though none explicitly employ an ORM that directly evaluates or optimizes the overall quality of an entire sequence of tool interactions.

## 2.3 Reward Modeling

Reward models (RMs) provide scalar preference signals that guide LLMs through preference optimization or RL (Wang et al., 2024). They can be broadly divided into ORMs (*outcome reward models*), which only evaluate the final output, and PRMs (*process reward models*), which score intermediate reasoning steps (Zhong et al., 2025). Early verifier-based approaches in the math domain (Cobbe et al., 2021) laid the foundation for ORMs, while later work explicitly contrasted outcome- and process-based supervision for math problems (Uesato et al., 2022), and developed PRMs that reward coherent stepwise reasoning (Lightman et al., 2023). Despite their promise, PRMs often face robustness and supervision challenges (Zhang et al., 2025), as highlighted by failed attempts reported by Guo et al. (2025). In contrast, ORMs have proven more scalable (Lin et al., 2025), focusing on final correctness and generalization, with recent advances such as Skywork-Reward (Liu et al., 2024a) demonstrating effective recipes for outcome-based training, achieving state-of-the-art performance on RewardBench (Lambert et al., 2024). Recently, tool-augmented reward models (Li et al., 2024), which enable reward models to utilize tools to produce a more accurate preference score, have been introduced. While prior work has studied RMs primarily in free-text reasoning and math/code domains, to the best of our knowledge, this is the first work introducing ORMs for tool calling, where a sequence of tool calls defines outcomes.

## 3 Methodology

### 3.1 FC-RewardBench Evaluation Dataset

While several benchmarks evaluate RMs on tasks involving chat, reasoning, safety (Lambert et al., 2024); factuality, instruction following, and math (Malik et al., 2025), there remains a notable gap in the evaluation of RMs for function-calling tasks. To bridge this gap, we propose FC-RewardBench, a benchmark specifically designed to evaluate RMs on function-calling tasks. This dataset comprises 1500 unique data points, each containing a user query, a tool catalog (tools available to the model to answer the user query), and the associated correct and incorrect tool calls for a given user query.

To construct FC-RewardBench, we utilize the single-turn splits of the BFCL-v3 dataset (Patil et al., 2025). The tool catalog, user query, and the correct tool calls in the dataset are directly sourced from BFCL-v3. Incorrect tool calls are generated using a pool of 25 language models, spanning sizes from 0.5B to 685B parameters. Each model is prompted to generate a tool call in response to the user query. The outputs are compared against the ground-truth, and only the incorrect generations are retained. From this pool, we randomly sample one incorrect call per instance to prevent over-representation from any single user query. Finally, 1,500 such examples are randomly selected to form the final dataset.

Table 1 presents a breakdown of error types observed in the dataset. Notably, a majority of the incorrect calls involve subtle errors such as incorrect parameter values, missing optional parameters, or an incorrect number of functions, which are non-trivial to detect. These characteristics require the RM to demonstrate a deeper understanding of the function-calling task, making FC-RewardBench a challenging and discriminative benchmark. Figure 2 shows a representative example from the

Table 1: Breakdown of errors in the FC-RewardBench dataset. The majority of errors in the dataset are subtle and hard to identify.

| Error Type | Count |
| --- | --- |
| Incorrect Parameter Value | 650 |
| Incorrect Function Name | 403 |
| Incorrect number of functions | 245 |
| Missing Optional Parameter | 78 |
| Missing Required Parameter | 45 |
| Incorrect Parameter Type | 43 |
| Unexpected Parameter | 21 |
| Incorrect output format | 15 |

Figure 2: Representative example from FC-RewardBench the parameter `player_count` is set to an incorrect value.

dataset, where the incorrect tool call sets the parameter `player_count` to an incorrect value. Additional details about the benchmark are provided in Appendix A.1.

## 3.2 REWARD MODELING

For pairwise preference modeling, RMs are commonly formulated using the Bradley–Terry model (Bradley & Terry, 1952), which defines the probability that output $y_+$ is preferred over $y_-$ given an input $x$ as:

$$p(y_+ \succ y_- \mid x) = \frac{\exp(r(x, y_+))}{\exp(r(x, y_+)) + \exp(r(x, y_-))} = \sigma(r(x, y_+) - r(x, y_-)) \quad (1)$$

where $r(x, y)$ is a scalar reward function, and $\sigma$ is the sigmoid function.

Training requires curating a dataset of pairwise preferences $D = \{(x, y_+, y_-) : y_+ \succ y_-\}$, with preferences obtained through either human annotations (Stiennon et al., 2020; Ouyang et al., 2022) or synthetic generation methods (Pace et al., 2024; Hosseini et al., 2024). The reward function $r$ is parameterized by a neural network $r_\theta$, typically initialized from a supervised fine-tuned model with the final layer replaced by a linear head.

The parameters of $r_\theta$ are estimated from the dataset $D$ using maximum likelihood estimation of the following objective:

$$J(r) = \max_{r_\theta} \mathbb{E}_{(x, y_+, y_-) \sim D}[log(\sigma(r_\theta(x, y_+) - r_\theta(x, y_-))] \quad (2)$$

In this work, we use reward centering (Eisenstein et al., 2023) to ensure that rewards are zero-centered. This is achieved by adding the following regularization term to the optimization objective:

$$J_{reg}(r) = J(r) + \eta \mathbb{E}_{(x, y_+, y_-) \sim D}[(r_\theta(x, y_+) + r_\theta(x, y_-))^2] \quad (3)$$

where $\eta$ is a small positive value hyperparameter.

## 3.3 TOOLRM TRAINING DATA GENERATION

To train ORMs for function-calling tasks, we require data consisting of user queries, tool catalogs, and the corresponding correct and incorrect tool calls. We construct this data by leveraging a diverse set of open-source, permissively licensed language models with function-calling capabilities. Specifically, we use publicly available function-calling datasets, which provide user queries, tool catalogs, and ground-truth tool call sequences. For each query, we prompt the models to generate tool calls using the tools specified in the dataset.

The generated tool calls are then compared against the ground-truth sequences. Outputs that deviate from the ground truth are retained as incorrect examples, while matching outputs are discarded.This procedure enables the collection of data that reflects the natural variability and error patterns of real-world models. It captures not only common mistakes but also subtle and complex failure modes that are difficult to anticipate or enumerate manually.

## 4 EXPERIMENTAL SETUP

**Training Data:**  To create training data for the RM, we select open-source datasets that cover various aspects of function-calling, such as the API-Gen dataset (Liu et al., 2024c) for single-turn interactions, the Schema-Guided Dialogue (SGD) dataset (Rastogi et al., 2020) for multi-turn interactions with tool invocations and responses, and the xlam-irrelevance[2] dataset for cases where the system lacks sufficient information to respond to a user query.

Since these datasets are common training datasets and our primary focus is to elicit representative incorrect behavior from the model, we follow Lin et al. (2024) and obfuscate the data samples to avoid the model regurgitating its training data. We obfuscate the samples by replacing function and parameter names with randomly generated strings and reordering the keys in the function schema.

We then use a collection of 11 permissively-licensed, open-weight models to generate the training data. The pool includes both general-purpose instruction-tuned models with function-calling capabilities and function-calling specific models, with parameter counts ranging from 0.5B to 32B. Specifically, we use the Qwen2.5-Instruct (Team, 2024) and Granite 3.3-Instruct (Granite Team) model series, along with Granite-20b-function-calling (Abdelaziz et al., 2024), SmolLM2 (Allal et al., 2025), Mistral-7b-Instruct-v0.3 and Mistral-Nemo-Instruct-2407.

After generating outputs from the model pool and keeping only the incorrect ones, we subsample one incorrect output per input user query to prevent over-representation from a user query in the training data. Overall, this results in 180,000 training data samples divided into 85,000 single and multi-turn data each, and 10,000 irrelevance data. The full list of models used to generate the training data, along with a few training data samples, is provided in Appendix A.2.

**Model architecture:**  We use the Qwen-2.5-Instruct models (Team, 2024; Yang et al., 2024) as the base architecture for our RMs. Specifically, we select the 1.5B, 7B, and 14B parameter variants, as they are Apache-2.0 licensed and offer a practical balance between size and performance. We initialize the RMs with the instruction-tuned model weights and replace the final language modeling head with a linear layer that maps the hidden representation to a scalar reward value.

The RMs accept the specifications of available functions, conversation history, and the generated tool call as input and produce a scalar reward as output (refer to Appendix A.3 for prompt template). We train all RMs for 1 epoch with a learning rate set to 1e-6, a cosine learning rate schedule with warmup set to 3% of total steps, and the reward centering coefficient set to 0.01.

**Benchmarks:**  In addition to FC-RewardBench, we evaluate models on the following commonly used function-calling benchmarks: Berkeley Function Calling Leaderboard (BFCL) v3 (Patil et al., 2025), API-Bank (Li et al., 2023), ToolAlpaca (Tang et al., 2023), NexusRaven API Evaluation [3], and SealTools (Wu et al., 2024). For API-Bank, we evaluate on the Call (API-Bank-1) and Retrieval+Call (API-Bank-2) splits. Table 2 summarizes their key statistics and characteristics. We highlight that these benchmarks vary in difficulty, encompassing single and multi-turn queries, nested tool calls, and evaluation sets collected from both real users and synthetically generated.

**Baselines:**  To evaluate performance on FC-RewardBench, we select eight RMs from *Reward-Bench*, spanning sizes from 3B to 70B parameters. We chose models that achieved high scores on RewardBench and support tool use in their chat template, which helps mitigate performance degradation due to prompt variability. In addition to these specialized RMs, we include six LLMs as judges, ranging from 70B to 685B parameters. See Appendix A.4 for the complete list of models.

For downstream task evaluations, we select the strongest function-calling models – the xLAM-2 series (Prabhakar et al., 2025a) – and the strongest generic instruction-tuned models – the Qwen3

---

[2]https://huggingface.co/datasets/MadeAgents/xlam-irrelevance-7.5k
[3]https://huggingface.co/datasets/Nexusflow/NexusRaven_API_evaluation

Table 2: Statistics of the evaluation benchmarks. "MT" denotes multi-turn queries.

| Dataset | # Examples | # Tools (avg./query) | # MT queries | Avg. MT turns | Nested calls | Avg. output tool calls | Data source |
|---|---|---|---|---|---|---|---|
| BFCL-v3 | 4,441 | 2,631 (3.3) | 800 | 4.2 | ✓ | 2.4 | Real |
| API-Bank | 473 | 64 (3.4) | 397 | 3.4 | ✗ | 1.0 | Real |
| ToolAlpaca | 100 | 64 (5.6) | 0 | – | ✗ | 1.5 | Synthetic |
| NexusRaven | 318 | 65 (7.4) | 0 | – | ✗ | 1.0 | Synthetic |
| SealTools | 627 | 3,036 (9.9) | 0 | – | ✓ | 2.9 | Synthetic |

series (Yang et al., 2025) – from the BFCL-v3 leaderboard. Both of these model series cover a wide range of sizes (0.6B to 70B), enabling a comprehensive assessment of ToolRM across model scales.

## 5 RESULTS

We evaluate our proposed RM to answer the following three research questions (RQ):

**RQ1:** How does ToolRM compare to existing RMs on FC-RewardBench?
**RQ2:** Can ToolRM improve the performance during inference through Best-of-$n$ sampling? And,
**RQ3:** Can ToolRM lead to better fine-tuned models through reward-guided data filtering?

### 5.1 RQ1: FC-REWARDBENCH EVALUATION

We evaluate ToolRM against state-of-the-art RMs from RewardBench (Lambert et al., 2024), Tool-Augmented RM (Themis) (Li et al., 2024), as well as leading LLMs used in an LLM-as-a-Judge setting, on the FC-RewardBench dataset.

RMs are evaluated by comparing scores assigned to the correct tool call outputs and incorrect tool call outputs for the same input. A prediction is counted as correct when the score for the correct tool call exceeds that of the incorrect one. LLMs-as-Judges are evaluated with a pairwise comparison prompt, where both candidate tool calls are presented and the model is instructed to select the correct one. To avoid position bias, the order of candidates is randomized. Experimental details, including the full prompt template, are provided in Appendix A.4. We show the results in Figure 1 and observe the following:

- **Specialized RMs under-perform on tool-calling tasks.** Despite strong performance in non-tool-calling domains, most specialized RMs fail to generalize effectively to the tool-calling domain. While some individual variants achieve higher scores, their performance remains inconsistent and generally below state-of-the-art levels. For example, the Tool-Augmented RM (Themis) attains only 45% accuracy on FC-RewardBench, highlighting its limited effectiveness in evaluating tool-calling behavior. Rule-based methods, such as Schema Validation, perform significantly worse than any learned model, underscoring the subtlety of errors in tool-calling tasks and the need for a learned reward model.
- **LLMs-as-Judges achieve higher accuracy but are computationally expensive.** LLMs-as-Judges attain strong performance on FC-RewardBench (exceeding 80% across all models), but their large parameter counts impose substantial computational costs.
- **ToolRM achieves the highest accuracy** on the benchmark while maintaining efficiency with respect to model size. The ToolRM-14B and ToolRM-7B variants outperform all other generative and sequential classifier models. Notably, even the ToolRM-1.5B variant surpasses the gpt-oss-120B model, approaching the performance of substantially larger Llama-4 models.

**Correlation with performance on downstream tasks:** The primary purpose of FC-RewardBench is to enable quick evaluation of RMs without having to do computationally expensive downstream evaluation. It is thus imperative that performance on FC-RewardBench reflects downstream task performance. To assess this, we select six generator models (Qwen3-1.7B, 8B, 32B, and xLAM-1B, 8b, 70B), 11 RMs (eight RMs from RewardBench and three ToolRM variants), and five benchmarks. For each generator model, RM, and dataset combination, we compute the perfor-

mance in a Best-of-$n$ ($n = 32$) setting and compute the Pearson correlation coefficient between the Best-of-$n$ performance and RM performance on FC-RewardBench. Results are shown in Figure 3.

Overall, we find that FC-RewardBench scores are strongly correlated with downstream task accuracy, with an average correlation of 0.84 across benchmarks and generator models. Across generator models, the average correlation ranges from 0.62 to 0.94, indicating that the alignment between FC-RewardBench and downstream performance is robust across model families. Importantly, this correlation remains stable even at scale: larger models such as Qwen3-32B and xLAM-2-70B continue to exhibit strong agreement between FC-RewardBench accuracy and downstream results. Taken together, these findings confirm that FC-RewardBench provides a reliable and computationally efficient proxy for expensive downstream evaluations.

## 5.2 RQ2: BEST-OF-$n$ SAMPLING WITH TOOLRM

In this section, we evaluate ToolRM in a Best-of-$n$ setting across multiple generator models. For each input, we sample $n = 32$ independent generations using temperature $T = 0.6$ from the generator model and use ToolRM to score and select the highest-ranked generation as the final output. Intuitively, a stronger RM should more reliably identify the correct tool call, thereby improving task performance. We compare against three baselines: Greedy Decoding, Majority Voting – where the most frequently occurring final answer is selected as the output, and Schema Validation – where we compare the output against the input tool schema and return the generation with the highest likelihood that validates the schema. For non-BFCL benchmarks, we report the Full Sequence Matching metric (Basu et al., 2025), which checks whether the predicted tool sequence – including tool names and argument-value pairs – exactly matches the gold sequence. For BFCL, we use its native evaluation metrics: AST-based scores for single-turn tasks and state-based/response-based metrics for multi-turn cases.

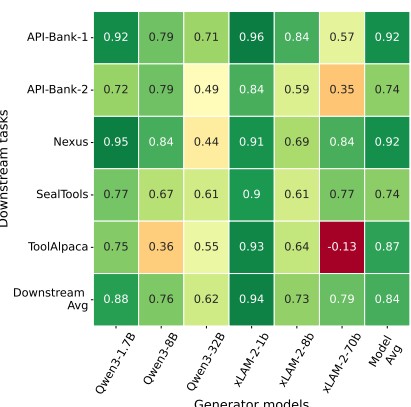

Figure 3: Correlation heatmap between FC-RewardBench performance and downstream accuracy across generator models and benchmarks, showing consistently strong alignment (avg. correlation = 0.84).

Figure 4 reports average performance across five benchmarks (API-Bank-1, API-Bank-2, ToolAlpaca, NexusRaven, and SealTools), while Table 3 presents results on the BFCL-v3 dataset. We summarize the key insights below.

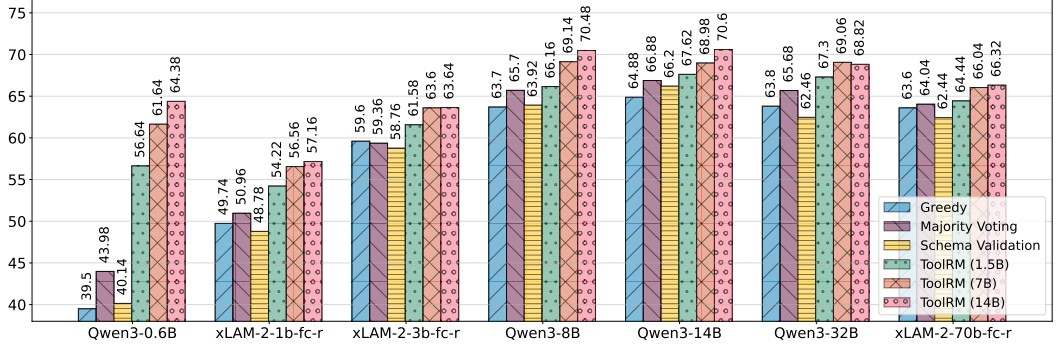

Figure 4: Performance of the Qwen3 series and xLAM-2 series in the Best-of-$n$ ($n = 32$) setting across five benchmarks: API-Bank-1, API-Bank-2, NexusRaven, ToolAlpaca, and SealTools.

- **Small Language Models (SLMs) benefit the most:** Best-of-$n$ sampling with Qwen3-0.6B and ToolRM-14B as the ranker improves accuracy from 39.5% to 64.38% – a gain of 24.9 points on non-BFCL benchmarks (Figure 4). This performance surpasses that of Qwen3-32B (63.8%) and Llama-xLAM-2-70b-fc-r (63.6%) with greedy decoding. On BFCL-v3, xLAM-2-1B-fc-r with ToolRM-14B improves overall, Non-Live AST, and Live AST accuracies by 3.2, 6.5, and 6.2

Table 3: Performance of the Qwen3 and xLAM-2 series of models in the Best-of-$n$ ($n = 32$) setting on BFCL-v3.

| Model | RM | Overall Acc | Non-Live AST | Live AST | Multi-Turn Acc |
|---|---|---|---|---|---|
| Qwen3-1.7b | Greedy | 55.74 | 80.23 | 71.35 | 10.25 |
| | Majority Voting | 57.96 | 83.90 | 74.24 | 10.12 |
| | Schema Validation | 56.34 | 83.48 | 72.46 | 8.25 |
| | ToolRM-14B | **61.05** | **89.79** | **80.01** | **14.12** |
| Qwen3-8b | Greedy | 64.65 | 88.90 | 80.09 | 26.38 |
| | Majority Voting | **67.96** | 90.33 | 81.72 | **33.13** |
| | Schema Validation | 67.21 | 90.58 | 81.05 | 32.50 |
| | ToolRM-14B | 67.14 | **92.19** | **82.98** | 31.50 |
| Qwen3-32b | Greedy | 69.19 | 89.33 | 82.83 | 38.25 |
| | Majority Voting | **73.57** | 91.38 | 83.64 | **46.63** |
| | Schema Validation | 69.38 | 89.71 | 82.61 | 37.75 |
| | ToolRM-14B | 70.61 | **92.31** | **84.23** | 39.62 |
| xLAM-2-1b-fc-r | Greedy | 54.09 | 68.98 | 54.77 | **35.12** |
| | Majority Voting | 53.51 | 69.42 | 54.92 | 31.50 |
| | Schema Validation | 54.21 | 70 | 55.51 | 33.88 |
| | ToolRM-14B | **57.28** | **75.50** | **60.92** | 34.25 |
| Llama-xLAM-2-8b-fc-r | Greedy | 71.14 | 84.31 | 67.80 | 67 |
| | Majority Voting | 72.39 | 84.90 | 67.75 | **67.75** |
| | Schema Validation | 70.89 | 84.79 | 66.47 | 65.38 |
| | ToolRM-14B | **72.52** | **87.73** | **72.46** | 61.62 |
| xLAM-2-32b-fc-r | Greedy | 76.34 | 89.40 | 75.35 | **65.25** |
| | Majority Voting | 76.28 | 89.38 | 75.43 | 64.12 |
| | Schema Validation | 75.90 | 89 | 74.76 | 64.13 |
| | ToolRM-14B | **76.54** | **90.27** | **77.42** | 63.25 |

points, respectively. Qwen3-1.7B achieves even larger improvements of 5.3, 9.6, and 8.7 points on these metrics (Table 3).

- **SLM + RM can match or surpass larger models:** Best-of-$n$ sampling with Qwen3-8B and ToolRM-14B improves non-BFCL benchmark accuracy by 6.8% points to 70.48%, which is 5.6 points higher than the best greedy baseline. On BFCL-v3, the same setup yields gains of 3.3 points on Non-Live AST and 2.9 points on Live AST, exceeding the performance of Qwen3-32B with greedy decoding.

- **Diminishing returns for very large models:** Improvements for large-scale generators (32B+) are modest. For instance, Llama-xLAM-2-32B-fc-r improves by only 2.1 points on non-BFCL benchmarks and 2.5 points on BFCL Live AST accuracy, suggesting limited additional utility of Best-of-$n$ sampling with very strong base models.

We also look at the breakdown of errors with greedy decoding and Best-of-$n$ sampling with ToolRM-14B, and present the results in Appendix A.5.

**Best-of-$n$ sampling improves model robustness:** We examine the impact of Best-of-$n$ sampling on model robustness to noise in the input. We utilize RoTBench (Ye et al., 2024), which comprises of 568 tool specifications and 105 user queries paired with tools with varying levels of noise. The *Clean* split contains tool and parameter names that clearly reflect their usage, while the *Slight*, *Medium*, and *Heavy* splits introduce increasing noise through operations such as character insertion and deletion, name reversal, and name swapping. The *Union* split combines all noisy variants and represents the most challenging setting. Model performance is evaluated across three tasks: Tool Selection, Parameter Identification, and Content Filling.

Table 4 reports results for greedy decoding and Best-of-$n$ ($n = 32$) with ToolRM-14B. We highlight two key findings. First, Best-of-$n$ decoding yields substantial gains across all models and tasks. For

Table 4: Performance of Qwen models on RoTBench with greedy decoding (Clean and Union) and Best-of-$n$ ($n = 32$) with ToolRM-14B (Clean@32 and Union@32).

| Generator Model | Tool Selection | | | | Parameter Identification | | | | Content Filling | | | |
|---|---|---|---|---|---|---|---|---|---|---|---|---|
| | Clean | Clean@32 | Union | Union@32 | Clean | Clean@32 | Union | Union@32 | Clean | Clean@32 | Union | Union@32 |
| Qwen-1.7B | 54.3 | **76.2** | 47.6 | *58.1* | 37.1 | **55.2** | 27.6 | *40.0* | 27.6 | **41.0** | 21.0 | *30.5* |
| Qwen-8B | 52.4 | **72.4** | 45.7 | *66.7* | 38.1 | **55.2** | 30.5 | *46.7* | 27.6 | **42.9** | 20.0 | *31.4* |
| Qwen-32B | 65.7 | **76.2** | 52.4 | *72.4* | 38.1 | **56.2** | 30.5 | *44.8* | 25.7 | **42.9** | 21.0 | *31.4* |

instance, Qwen-8B improves Tool Selection from 52.4 to 72.4 on the Clean split, while performance on the Union split rises from 45.7 to 66.7. Comparable gains of 15–25 points are observed for Parameter Identification and Content Filling. Second, Union@32 consistently outperforms the Clean baseline, despite Union being the more difficult split. For example, Qwen-32B achieves 72.4 on Tool Selection under Union@32 compared to 65.7 under Clean, showing that Best-of-$n$ decoding not only mitigates noise but can also exceed performance on noise-free data.

## 5.3 RQ3: REWARD-GUIDED FINE-TUNING

### 5.3.1 TOOLRM FOR DATA FILTERING

In this experiment, we assess the effectiveness of using ToolRM as a data filter to construct a high-quality training dataset for tool-use models. We curate a training corpus comprising both single-turn and multi-turn examples drawn from APIGen-MT (Liu et al., 2024c), SealTools (Wu et al., 2024), Glaive V2[4], and Granite function-calling dataset (Abdelaziz et al., 2024), yielding a total of 16K samples. We highlight that these datasets have no overlap with ToolRM training data, thus allowing us to test the generalization capabilities of ToolRM. We select Llama-3.1-8B-Instruct (Grattafiori et al., 2024) as the base model and performed LoRA-based fine-tuning (Hu et al., 2022) to train each variant for 1 epoch with a learning rate of 2e-4, a LoRA rank of 16, alpha of 32, a cosine scheduler, and a warmup ratio of 10%.

Table 5: Finetuning results of Llama-3.1-8B-Instruct on three training subsets: full 16K dataset (FT-16K), 8K randomly sampled (FT-Random-8K), and top 8K selected by ToolRM-14B (FT-Best-8K).

| Llama-3.1-8B-Instruct | BFCL V3 | ToolAlpaca | Nexus | API-Bank-1 | API-Bank-2 | Sealtools | AVG |
|---|---|---|---|---|---|---|---|
| Base | 49.6 | 38.0 | 64.8 | **67.9** | **66.2** | 37.6 | 54.0 |
| FT-16K | 54.1 | 43.0 | **75.5** | 57.4 | 63.5 | 72.7 | 61.0 |
| FT-Random-8K | 55.2 | **44.0** | 74.2 | 49.9 | 54.1 | 73.2 | 58.4 |
| FT-Best-8K | **55.4** | **44.0** | 72.0 | 63.7 | **66.2** | **73.7** | **62.5** |

Table 5 compares the performance of the base model with three fine-tuned variants: (1) trained on the full 16K dataset (FT-16K), (2) trained on a random 8K subset (FT-Random-8K), and (3) trained on the top 8K samples as ranked by ToolRM-14B (FT-Best-8K). We highlight the following key insights from the results.

- **Fine-tuning consistently improves performance.** All fine-tuned models outperform the base model, raising average accuracy from 54.0% to 61.0% when trained on the full dataset.
- **Naive subsampling degrades performance.** Training on a random 8K subset reduces accuracy to 58.4% – 2.6 points below the model fine-tuned on the entire corpus. This performance degradation reflects the inapplicability of naively subsampling to reduce dataset size since it results in both the inclusion of low-quality samples and the exclusion of high-quality ones.
- **ToolRM based data filtering achieves the best performance.** Selecting the top 50% of samples with ToolRM-14B yields the strongest results at 62.5%, surpassing the full-data model while using only half the training corpus. This demonstrates that ToolRM can effectively identify high-quality data, enabling superior performance under tighter data budgets.

These results highlight the importance of data quality in fine-tuning tool-use models and show that reward-guided filtering of low-quality data can yield superior performance with less training.

### 5.3.2 POLICY OPTIMIZATION USING TOOLRM

To assess the utility of ToolRM for policy optimization, we follow ToolRL (Qian et al., 2025a) and train models using Group Relative Policy Optimization (GRPO) (Shao et al., 2024). ToolRL defines the reward as $R = R_{\text{format}} + R_{\text{correctness}}$, where $R_{\text{format}} \in \{0, 1\}$ evaluates whether the output adheres to the format specified in the prompt, and $R_{\text{correctness}}$ measures the correctness of the tool-call output. We consider three variants of $R_{\text{correctness}}$: (1) $R_{\text{schema}} \in \{-1, 1\}$, validates the predicted tool calls against the provided tool specifications, assigning $-1$ if there are schema violations and $+1$ otherwise; (2) $R_{\text{ToolRL}} \in [-3, 3]$, follows Qian et al. (2025a) and computes rewards by comparing

---

[4]https://huggingface.co/datasets/glaiveai/glaive-function-calling-v2

Table 6: BFCL-v3 performance of Llama-3.2-3B-Instruct and Qwen2.5-3B-Instruct trained with GRPO under three reward designs.

| Model | Reward Variant | Non-Live AST Acc | Live AST Acc |
|-------|---------------|------------------|--------------|
| Llama-3.2-3B-Instruct | Base | 15.35% | 43.82% |
| | $R_{\text{Schema}}$ | 51.71% | 62.25% |
| | $R_{\text{ToolRL}}$ | 75.27% | 64.25% |
| | $R_{\text{ToolRM}}$ | **78.40%** | **64.32%** |
| Qwen2.5-3B-Instruct | Base | 43.06% | 55.66% |
| | $R_{\text{Schema}}$ | 63.17% | 66.54% |
| | $R_{\text{ToolRL}}$ | **80.42%** | 67.21% |
| | $R_{\text{ToolRM}}$ | 79.58% | **67.51%** |

Table 7: Ablation results for ToolRM-1.5B model on FC-RewardBench dataset.

| Model | Accuracy |
|-------|----------|
| ToolRM-1.5B | 81.88% |
| **Hyperparameter Ablation** | |
| Without obfuscation | 68.25% |
| High reward centering ($\eta = 0.1$) | 80.02% |
| No reward centering ($\eta = 0$) | 80.32% |
| **Data Ablation** | |
| Without API-Gen | 65.48% |
| Without SGD | 81.03% |
| **Generator model ablation** | |
| Only large ($>=12B$) models | 78.70% |
| Only small ($<=2B$) models | 81.01% |

predicted and ground-truth tool calls; and (3) $R_{\text{ToolRM}} \in [-3, 3]$, scores the tool calls using ToolRM-14B. Notably, $R_{\text{schema}}$ and $R_{\text{ToolRM}}$ do not require access to ground-truth tool calls, making them more appropriate for RL settings, whereas $R_{\text{ToolRL}}$ requires ground truth, limiting its applicability.

We train Llama-3.2-3B-Instruct and Qwen2.5-3B-Instruct models using the three reward variants and evaluate them on the BFCL-v3 dataset, with results shown in Table 6. Across both models, all three reward variants substantially improve performance over the base models. Simple schema-based rewards provide strong gains without requiring ground-truth supervision, yielding average improvements of about 20 points. ToolRM-based rewards achieve the best overall results: for Llama-3.2-3B-Instruct, $R_{\text{ToolRM}}$ attains the highest accuracy on both evaluation metrics, and for Qwen2.5-3B-Instruct, it provides the best Live AST accuracy, surpassing the gold-dependent $R_{\text{ToolRL}}$. Overall, $R_{\text{ToolRM}}$ consistently matches or exceeds $R_{\text{ToolRL}}$ despite not requiring access to ground truth, highlighting its practicality and effectiveness as a scalable reward signal for reinforcement learning in the tool-calling setting. Additional experimental details are provided in Appendix A.7.

# 6 ABLATION ANALYSIS

We conduct an ablation study to assess the contribution of individual components to the overall performance of ToolRM. Specifically, we train ToolRM-1.5B, ablating different hyperparameters, training datasets, and generator models, and evaluating each variant on the FC-RewardBench dataset. Table 7 summarizes the results. First, we observe that the full ToolRM-1.5B model, incorporating all components, achieves the highest performance. Second, removing the API-Gen dataset leads to a 16.4-point drop in performance, underscoring its significance in training. Finally, obfuscating tool and parameter names results in a 13.63-point reduction in performance. This suggests that obfuscation prevents the model from overfitting to specific tool or parameter names and encourages it to attend to other parts of the tool specifications, thereby improving robustness and generalization.

# 7 CONCLUSION

In this paper, we presented a comprehensive framework for reward modeling in tool-calling scenarios, addressing a critical gap in current LLM evaluation pipelines. Our benchmark, FC-RewardBench, enables systematic assessment of reward models on tool-based reasoning. We also presented a framework for training outcome RMs that outperform existing significantly larger RMs in the tool calling setting. When used for inference-time scaling, ToolRM improves the performance of general-purpose and tool calling models by up to 25% on various tool calling benchmarks. Looking ahead, we see several promising directions for advancing reward modeling in this domain. First, moving beyond classification-based RMs to generative verifiers with chain-of-thought reasoning could improve robustness and interpretability. Second, incorporating the tool and environment state into training could help models safely recover from execution failures. Finally, bridging outcome and process reward modeling may offer a unified framework that balances scalability with fine-grained control over reasoning quality.

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

# A APPENDIX

## A.1 FC-REWARDBENCH BENCHMARK DETAILS

The list of models included in FC-RewardBench, along with the number of incorrect tool call output samples per model is provided in Table 8.

| Model Name | Count |
|---|---|
| Qwen/Qwen2.5-0.5B-Instruct | 450 |
| Qwen/Qwen2.5-0.5B-Instruct-FC | 237 |
| ibm-granite/granite-20b-functioncalling | 112 |
| Qwen/Qwen2.5-1.5B-Instruct | 102 |
| BitAgent/BitAgent-8B | 74 |
| DeepSeek-R1 | 64 |
| openbmb/MiniCPM3-4B-FC | 59 |
| NovaSky-AI/Sky-T1-32B-Preview | 54 |
| Qwen/Qwen2.5-1.5B-Instruct-FC | 52 |
| speakleash/Bielik-11B-v2.3-Instruct | 41 |
| Qwen/Qwen2.5-14B-Instruct-FC | 38 |
| openbmb/MiniCPM3-4B | 38 |
| Qwen/Qwen2.5-14B-Instruct | 28 |
| Qwen/Qwen2.5-7B-Instruct | 23 |
| ZJared/Haha-7B | 22 |
| meetkai/functionary-small-v3.1-FC | 21 |
| watt-ai/watt-tool-70B | 21 |
| Qwen/Qwen2.5-7B-Instruct-FC | 18 |
| Qwen/Qwen2.5-32B-Instruct-FC | 15 |
| Qwen/Qwen2.5-32B-Instruct | 13 |
| meetkai/functionary-medium-v3.1-FC | 11 |
| Team-ACE/ToolACE-2-8B | 6 |
| Qwen/QwQ-32B-Preview | 1 |

Table 8: Breakdown of errors by models in FC-RewardBench

## A.2 TOOLRM TRAINING DATA DETAILS

We use the following models to generate the training data for ToolRM:

- `ibm-granite/granite-3.3-2b-instruct`
- `ibm-granite/granite-3.3-8b-instruct`
- `ibm-granite/granite-20b-functioncalling`
- `HuggingFaceTB/SmolLM2-1.7B-Instruct`
- `Qwen/Qwen2.5-0.5B-Instruct`
- `Qwen/Qwen2.5-1.5B-Instruct`
- `Qwen/Qwen2.5-7B-Instruct`
- `Qwen/Qwen2.5-14B-Instruct`
- `Qwen/Qwen2.5-32B-Instruct`
- `mistralai/Mistral-7B-Instruct-v0.3`
- `mistralai/Mistral-Nemo-Instruct-2407`

A few samples from the training data are shown in Figure 5.

## A.3 TOOLRM PROMPT

The prompt used to train the ToolRM is shown in Listing 1.

```
Tool Catalog                                    User Query

[                                               Fetch the details of a vehicle with license plate number 'XYZ456'. Also,
  {                                             get the horoscope information for a person born under the Scorpio sign.
    "name": "RMof3S1",
    "description": "Fetch horoscope information for a
given astrological sign using the Horoscope Astrology      Correct Tool Call          Incorrect Tool Call
API.",
    "parameters": {                             [                              [
      "NZImuPDflZg": {                             {                              {
        "description": "The astrological sign to fetch         "name": "6nH2QvU",             "name": "6nH2QvU",
information for. Valid options include 'aries',               "arguments": {                 "arguments": {
'taurus', 'gemini', 'cancer', 'leo', 'virgo', 'libra',         "qTVPjX6JtO": "XYZ456"          "qTVPjX6JtO": "XYZ456"
'scorpio', 'sagittarius', 'capricorn', 'aquarius', and      }                              }
'pisces'.",                                     },                             }
        "default": "libra",                      {                            ]
        "type": "str"                              "name": "RMof3S1",
      }                                            "arguments": {
    }                                                "NZImuPDflZg": "scorpio"
  },                                             }
  {                                           }
    "name": "6nH2QvU",                         ]
    "description": "Fetch vehicle information from the
Mexican Vehicle Registry using the provided license
plate number and optional RapidAPI key.",
    "parameters": {
      "qTVPjX6JtO": {
        "default": "Y20BBG",
        "description": "The license plate number for
which to retrieve the vehicle information.",
        "type": "str"
      }
    }
  }
]
```

```
Tool Catalog                                    User Query

[                                               Can you retrieve the last 5 head-to-head encounters for the football match
  {                                             with ID 345678?
    "name": "pmSfQnDtVmP",
    "description": "Fetches and returns head-to-
head statistics and previous encounters for the home       Correct Tool Call          Incorrect Tool Call
and away team of an upcoming match.",
    "parameters": {                             [                              [
      "iBWXe7XXX": {                               {                              {
        "default": "81930",                           "name": "pmSfQnDtVmP",         "name": "pmSfQnDtVmP",
        "type": "int",                                "arguments": {                 "arguments": {
        "description": "The ID of the match to             "bmjcV0el7qNbk": 5,            "iBWXe7XXX": 345678
get statistics for."                                       "iBWXe7XXX": 345678          }
      },                                            }                            }
      "bmjcV0el7qNbk": {                          }                            ]
        "description": "Limits the search to      }
only X previous encounters. The default is 10, with a    ]
maximum of 10.",
        "type": "int, optional",
        "default": "10"
      }
    }
  }
]
```

Figure 5: Data samples from ToolRM training data. Each sample has a tool catalog, a conversation between the user and assistant, along with the corresponding correct and incorrect tool calls. The top sample is missing one tool call from the Incorrect version, while the Bottom sample is missing a parameter from the tool call.

```
1  <|im_start|>system
2  You are provided with a user query, a catalog of tools available to fulfill that user query, and a list
       ↪ of tool calls that use tools available in the catalog to fulfill user request.
3  Your job is to assess whether the tool calls adequately fulfill the user request or not.
4
5  You have the following tools available:
6
7  ```json
8  [
9          {"name": "diabetes_prediction", "description": "Predict the likelihood of diabetes type 2
              ↪ based on a person's weight and height.", "parameters": {"type": "dict", "
              ↪ properties": {"weight": {"type": "integer", "description": "Weight of the person
              ↪ in lbs."}, "height": {"type": "integer", "description": "Height of the person in
              ↪ inches."}, "activity_level": {"type": "string", "enum": ["sedentary", "lightly
              ↪ active", "moderately active", "very active", "extra active"], "description": "
              ↪ Physical activity level of the person."}}, "required": ["weight", "height", "
              ↪ activity_level"]}}
10 ]
11 ```
12
13 <|im_end|>
14 <|im_start|>user
15 Predict whether a person with weight 150lbs and height 5ft 10in who is lightly active will get type
       ↪ 2 diabetes.<|im_end|>
16 <|im_start|>assistant
17 ```json
18 [
19         {"diabetes_prediction": {"weight": 150, "height": 68, "activity_level": "lightly active"}}
20 ]
21 ```<|im_end|>
```

Listing 1: ToolRM prompt

| Error type | Greedy | ToolRM-14B |
|---|---|---|
| Incorrect Parameter Value | 321 | 231 |
| Irrelevance error | 185 | 210 |
| Malformed output syntax | 86 | 34 |
| Incorrect function name | 62 | 39 |
| Missing optional parameter | 36 | 26 |
| Incorrect parameter type | 31 | 24 |
| Wrong number of functions | 21 | 9 |
| Total | 742 | 573 |

Table 9: Breakdown of errors with the Qwen3-1.7B model as generator with Greedy decoding and Best-of-32 sampling with ToolRM-14B

### A.4 FC-REWARDBENCH EXPERIMENT DETAILS

#### A.4.1 MODEL DETAILS

We include the following models from RewardBench:

- `Ray2333/GRM-Llama3.2-3B-rewardmodel-ft`
- `Skywork/Skywork-Reward-V2-Qwen3-4B`
- `Skywork/Skywork-Reward-V2-Qwen3-8B`
- `LxzGordon/URM-LLaMa-3.1-8B`
- `Skywork/Skywork-Reward-Llama-3.1-8B-v0.2`
- `nicolinho/QRM-Llama3.1-8B-v2`
- `infly/INF-ORM-Llama3.1-70B`
- `Skywork/Skywork-Critic-Llama-3.1-70B`

We include the following LLMs as Judges:

- `deepseek-ai/DeepSeek-V3`
- `meta-llama/Llama-3.1-405B-Instruct-FP8`
- `meta-llama/Llama-3.3-70B-Instruct`
- `meta-llama/Llama-4-Scout-17B-16E`
- `meta-llama/Llama-4-Maverick-17B-128E-Instruct`
- `openai/gpt-oss-120b`

Additionally, we include the Tool-Augmented Reward Model (`ernie-research/Themis-7b`).

#### A.4.2 LLM-AS-JUDGE PROMPT

The prompt used to evaluate LLMs-as-Judges on FC-RewardBench is shown in Listing 2. The placeholders {`tool-library`}, {`query`}, {`response-A`}, and {`response-B`} are replaced with appropriate values.

### A.5 ERROR ANALYSIS

In this experiment, we evaluate the impact of ToolRM-14B on error reduction in the Best-of-$n$ ($n = 32$) sampling setting, using the Qwen3-1.7B generator on the single-turn splits of BFCL-v3. As reported in Table 9, ToolRM-14B decreases the total error count from 742 to 573, corresponding to a 22.7% relative reduction. The predominant error type, Incorrect Parameter Value, responsible for nearly 42% of greedy decoding errors, is reduced by 28%, indicating that ToolRM is particularly effective at mitigating semantic mis-specification of parameter values. Moreover, errors such as Incorrect Function Name and Wrong Number of Functions are reduced by 37% and 57%, respectively. In contrast, Irrelevance Errors increase from 185 to 210, suggesting that ToolRM tends to favor producing function call outputs even in cases where no valid call can appropriately satisfy the user query.

```
1   Please act as an impartial judge and evaluate the quality of the responses provided by two AI
        ↪  assistants to the user question displayed below. You should choose the assistant that
        ↪  follows the user's instructions and answers the user's question best.
2
3   You will be given:
4   − TOOL SPECIFICATIONS: All the tool specifications including parameters and their
        ↪  descriptions available to the assistant to answer the query.
5   − CONVERSATION: Conversation between the user and the assistant
6   − ASSISTANT RESPONSES: List of responses from two assistants − [[A]] and [[B]]. Each
        ↪  response is a sequence of tool calls needed to answer the question.
7
8   When comparing two tool call sequences for the same user query, carefully evaluate both
        ↪  sequences and determine which one better follows the tool specifications and the question
        ↪  requirements.
9
10  Consider the following instructions to compare the assistant responses:
11  − Check whether all the tools used are relevant and actually exist.
12  − Verify that the tools are called in a correct and logical order.
13  − Ensure that the correct parameters are used for each tool and that no nonexistent parameters are
        ↪  included.
14  − Confirm that parameter values and formats are appropriate based on the question and tool
        ↪  specifications.
15  − Make sure all required parameters and data mentioned in the question are included.
16  − Look for any extra tools or parameters that are not needed or mentioned in the question.
17
18  Also, follow these general instructions:
19  − Begin your evaluation by comparing the two responses (i.e., [[A]] and [[B]]) and provide a short
        ↪  explanation.
20  − Avoid any position biases and ensure that the order in which the responses were presented does
        ↪  not influence your decision.
21  − Do not allow the length of the responses to influence your evaluation.
22  − Do not favor certain names of the assistants.
23  − Be as objective as possible.
24  − After providing your explanation, output your final verdict by strictly following this format: "[[A
        ↪  ]]" if assistant A's response is better and "[[B]]" if assistant B's response is better.
25  − STRICTLY follow this output format: [EXPLANATION]\n{Short comparison highlighting
        ↪  differences and reasoning.}\n\n[VERDICT]\n{[[A]] or [[B]]}. Place your reasoning
        ↪  after the [EXPLANATION] tag and your final choice after the [VERDICT] tag.
26
27  # TOOL SPECIFICATIONS
28  {tool−library}
29
30  # CONVERSATION
31  {query}
32
33  # ASSISTANT RESPONSES:
34  [[A]] {response−A}
35  [[B]] {response−B}
36
37  # ANSWER:
```

Listing 2: LLM-as-Judge Prompt used for FC-RewardBench evaluation

### A.6 TOOLRM GENERALIZATION

To assess how well ToolRM generalizes to inputs unseen during training, we embed each tool and conversation from both the training set and the five non-BFCL test sets using the `all-mpnet-base-v2`[5] embedding model. We then compute, for each test example, the maximum cosine similarity with the training set and plot the resulting distributions in Figure 6. We observe low similarity between training and test examples for both tools and conversations (median $< 0.6$ in both cases), indicating that ToolRM generalizes effectively to novel inputs at test time.

### A.7 GRPO EXPERIMENTAL DETAILS

We utilize the open-source library `verl`[6] and the code base and dataset provided by ToolRL [7] to train models for GRPO experiments. The hyperparameters used to train the model are listed in Table 10, and the prompt used to train the models is listed in Listing 3.

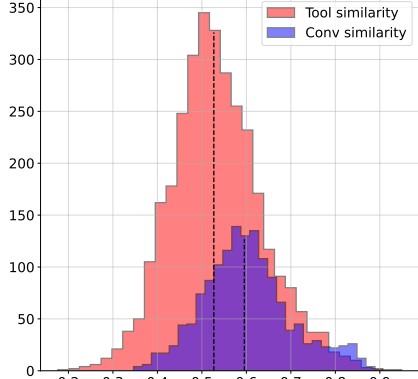

Figure 6: Histogram of maximum cosine similarity between train and test tools and conversations.

| Hyperparameter | Value | Hyperparameter | Value |
|---|---|---|---|
| Max prompt length | 2048 | Number of rollouts | 4 |
| Max response length | 1024 | Total epochs | 15 |
| Learning rate | 1e-6 | Train batch size | 512 |
| PPO mini batch size | 128 | Validation batch size | 128 |

Table 10: GRPO Experimental Hyperparameters

---

[5]https://huggingface.co/sentence-transformers/all-mpnet-base-v2

[6]https://github.com/volcengine/verl

[7]https://github.com/qiancheng0/ToolRL

```
1   You are a helpful multi−turn dialogue assistant capable of leveraging tool calls to solve user tasks
        ↪ and provide structured chat responses.
2
3   **Available Tools**
4   ```json
5   {{TOOLS}}
6   ```
7
8   **Steps for Each Turn**
9   1. **Think:** Recall relevant context and analyze the current user goal.
10  2. **Decide on Tool Usage:** If a tool is needed, specify the tool and its parameters.
11  3. **Respond Appropriately:** If a response is needed, generate one while maintaining
        ↪ consistency across user queries.
12
13  **Output Format**
14  ```plaintext
15  <think> Your thoughts and reasoning </think>
16  <tool_call>
17  {"name": "Tool name", "parameters": {"Parameter name": "Parameter content", "... ...": "... ..."}}
18  {"name": "... ...", "parameters": {"... ...": "... ...", "... ...": "... ..."}}
19  ...
20  </tool_call>
21  <response> AI's final response </response>
22  ```
23
24  **Important Notes**
25  1. You must always include the '<think>' field to outline your reasoning. Provide at least one of
        ↪ '<tool_call>' or '<response>'. Decide whether to use '<tool_call>' (possibly multiple
        ↪ times), '<response>', or both.
26  2. You can invoke multiple tool calls simultaneously in the '<tool_call>' fields. Each tool call
        ↪ should be a JSON object with a "name" field and an "parameters" field containing a
        ↪ dictionary of parameters. If no parameters are needed, leave the "parameters" field an
        ↪ empty dictionary.
27  3. Refer to the previous dialogue records in the history, including the user's queries, previous '<
        ↪ tool_call>', '<response>', and any tool feedback noted as '<obs>' (if exists).
```

Listing 3: Prompt used for GRPO training of models

