# OpenReview forum: "ToolRM: Outcome Reward Models for Tool-Calling Large Language Models"
_ICLR.cc/2026/Conference — ICLR 2026 Conference Withdrawn Submission_

### Official Review · Reviewer_p6AE · 2025-10-31

**Soundness:** 3
**Presentation:** 3
**Contribution:** 3
**Rating:** 2
**Confidence:** 3

**Summary:**

This paper focuses on reward modeling for tool-calling in LLMs. The authors introduce FC-RewardBench, a benchmark explicitly designed to evaluate outcome reward models (ORMs) in function and tool-calling contexts. They propose ToolRM, a new family of ORMs trained with data synthesized from open-weight LLMs, with experiments demonstrating that ToolRM offers significant improvements—up to 25% average gains—across multiple tool-use benchmarks. Additional analyses highlight ToolRM’s effectiveness for data filtering and increased robustness, especially in low-data regimes and noisy conditions.

**Strengths:**

- Reward modeling for tool use is an area that remains relatively unaddressed in existing reward modeling literature.
- The paper is relatively clearly written.

**Weaknesses:**

- The motivation for training a reward model for tool use is unclear. If tool use and task performance (e.g., QA), in both the training and inference settings, are verifiable, what is the motivation for training a reward model to evaluate the effectiveness of tool use?
- The benchmark is mostly adapted from other open source benchmarks (specifically, BFCL-v3). The training set is made by constructing negative examples. Other than this, the training paradigm for ToolRM is well established by previous work. These made the contribution of the submission limited.
- The experiments show that the performance of ToolRM underperforms inference scaling methods like majority voting or vanilla model with greedy decoding. This possibly demonstrates the ineffectiveness of the training set constructed in the paper.

**Questions:**

- How sensitive is ToolRM’s performance to the choice of the reward centering coefficient $\eta$ and to architectural or regularization hyperparameters? Can an ablation be provided?
- How generalizable is ToolRM to entirely new tool schema (e.g., tools unseen during training or with radically different parameter signatures) or domains outside API calls?
- The error analysis (Table 7) shows some error types (e.g., irrelevance errors) increase with ToolRM. What caused this?

---

> ### Author Response · Authors · 2025-11-24
> **Author response to Reviewer p6AE**
>
> We thank the reviewer for taking the time to review our work and provide valuable critical feedback. We are encouraged that you found reward modeling for tool-use an underexplored area, and the paper to be well-written. We address each of your concerns in detail below and have incorporated the corresponding improvements into the revised manuscript.
>
> - **Ablation Study**
>     - Thank you for the suggestion. We have **added an ablation study in Section 6** of the updated manuscript (**results also provided in the next comment**).
>     - We report ToolRM's performance by varying the hyperparameters (reward centering coefficient, without obfuscation), training datasets (without API-Gen, and SGD), and generator models
>     - We observe that excluding the API-Gen training dataset and Obfuscation has the largest impact on performance
>
> - **ToolRM Generalizability**
>     - We have **added a new section in Appendix A.6** that directly evaluates how well ToolRM handles unseen tool schemas.
>     - As summarized in that section, we find that tools and conversations in the test set have low similarity to the training set (median < 0.6), yet ToolRM continues to perform reliably, indicating strong generalization to new tool types and parameter signatures.
>     - We kindly refer the reviewer to this section for full details of the analysis.
>
> - **Motivation for training a reward model is unclear**
>     - We clarify that while tool use allows for syntactic verification of generated tool calls against input tool schema (correct parameter types, required params included, no hallucinations, etc), full semantic verifiability (correct parameter values, no unnecessary or missing tool calls) is unattainable during test time as ground-truth labels are unavailable.
>     - Our experiments demonstrate that relying solely on partial verifiability (schema validation) severely underperforms learned reward models (**Figure 1 and Figure 4**). This serves as the motivation to train a reward model to serve as a robust proxy for correctness during inference (e.g., Best-of-N) and RL exploration where gold-truths are absent.
>     - We have updated the manuscript to make this point clear. Thank you for highlighting this to us.
>
> - **Limited contributions**
>     - Our work introduces several novel and impactful components:
>         - FC-RewardBench: The first benchmark for evaluating reward models in tool-calling settings, with strong correlation to downstream task performance.
>         - Training Framework: A method for building an ORM for tool-calling using data generated from permissively licensed, open-weight LLMs.
>         - ToolRM Models: Multiple reward models (1.7B–14B parameters) trained and extensively evaluated on seven out-of-domain benchmarks, demonstrating generalization.
>         - Comprehensive Evaluation: Comparison against greedy decoding, majority voting, and schema validation in Best-of-n (n=32) settings across diverse tool-calling models, achieving up to 25% average improvement and enabling efficient data filtering for better fine-tuned models with less data.
>     - These contributions collectively advance the state of reward modeling for tool-calling, addressing a previously unexplored challenge and providing practical benefits for real-world deployments.
>
> - **ToolRM underperforms inference scaling methods**
>     - Thank you for your observation. We would like to clarify that ToolRM consistently outperformed inference scaling methods such as majority voting, schema validation, and greedy decoding in API-Bank-1, API-Bank-2, NexusRaven, ToolAlpaca, and SealTools (as shown in Figure 4) by a significant margin. For smaller models, the difference between Majority Voting and ToolRM-14B performance is as big as 20 points, and even for larger models, the performance difference is 2-3 points.
>     - For BFCL v3, ToolRM achieves the best results in 4 out of the 6 models evaluated. In the remaining two cases, the differences are marginal: for Qwen3-8B, ToolRM scores 67.14 vs. 67.96 for majority voting, and for Qwen3-32B, ToolRM scores 70.61 vs. 73.57. These results demonstrate that ToolRM is competitive even against inference scaling approaches, and in most scenarios, it provides superior performance.
>
>
> We hope these revisions and clarifications fully address your concerns. We would be grateful for your reconsideration of the overall rating.

---

> > ### Author Response · Authors · 2025-11-24
> >
> > Ablation results for ToolRM-1.5B model on FC-RewardBench dataset.
> >
> > |                Model               | Accuracy |
> > |:----------------------------------:|:--------:|
> > |             ToolRM-1.5B            |  81.88%  |
> > |       **Hyperparameter Ablation**             |
> > |         Without obfuscation        |  68.25%  |
> > | High reward centering ($\eta=0.1$) |  80.02%  |
> > |   No reward centering ($\eta=0$)   |  80.32%  |
> > |            **Data Ablation**                  |
> > |           Without API-Gen          |  65.48%  |
> > |             Without SGD            |  81.03%  |
> > |      **Generator model ablation**             |
> > |     Only large ($>=$12B) models    |  78.70%  |
> > |     Only small ($<=$2B) models     |  81.01%  |

---

> > > ### Author Response · Authors · 2025-11-26
> > >
> > > Dear Reviewer,
> > >
> > > We would be very glad to engage in discussion, so please feel free to reach out with any questions or points that may benefit from further clarification. Thank you again for your time and thoughtful evaluation.

---

### Official Review · Reviewer_3mfp · 2025-11-01

**Soundness:** 3
**Presentation:** 2
**Contribution:** 2
**Rating:** 4
**Confidence:** 3

**Summary:**

This paper focuses on reward modeling for tool-based reasoning scenarios. The authors first propose FC-RewardBench, a benchmark for systematic evaluation of reward models in tool-calling contexts. Subsequently, they introduce a training framework for outcome reward models. Experimental results demonstrate the effectiveness of their approach and highlight the importance of reward modeling in tool-calling reasoning scenarios.

**Strengths:**

1. The paper is well-structured with clear writing that facilitates comprehension of the proposed methodology.
2. The proposed FC-RewardBench represents a significant contribution to the tool-calling domain, providing a systematic evaluation framework for reward models that will facilitate the development of reward modeling in this scenario.
3. The authors conduct experiments on FC-RewardBench and multiple datasets, demonstrating the importance of reward modeling. Additionally, they evaluate the proposed ToolRM using best-of-n sampling, showing consistently strong performance.

**Weaknesses:**

1. In constructing FC-RewardBench, correct tool calls are sourced from original data while incorrect tool calls are generated by various models. This approach may introduce structural or pattern-based differences between correct and incorrect tool calls, or result in errors that have minimal or negligible impact on the overall reasoning outcome. The paper lacks effective mechanisms to manage these potential biases.
2. For tool-calling reasoning, many scenarios can be constrained through rule-based outcome rewards. Moreover, process rewards for constraining multi-turn tool calls are equally important. However, the paper exclusively focuses on outcome rewards without discussing process reward modeling approaches.
3. The experimental evaluation lacks direct application of the reward models to downstream tasks, such as demonstrating the enhancement effects on tool-calling agents through integration with PPO or GRPO training procedures.

**Questions:**

N/A

---

> ### Author Response · Authors · 2025-11-24
> **Author response to Reviewer 3mfp**
>
> We thank the reviewer for taking the time to review our work and provide valuable critical feedback. We are encouraged that you found FC-RewardBench a significant contribution, our experiments comprehensive, and our empirical results strong. We address each of your concerns in detail below and have incorporated the corresponding improvements into the revised manuscript.
>
>
> - **Biases in FC-RewardBench**
>   - Thank you for highlighting this important point. We agree that benchmark construction requires careful attention to potential biases. Several aspects of our methodology help mitigate these concerns:
>     - Diversity of error sources: FC-RewardBench uses incorrect calls from 25 different models spanning a wide range of architectures, training regimes, and model sizes. This diversity helps avoid systematic or uniform structural differences in the resulting dataset.
>     - Balanced error-type distribution: As shown in Table 1, the dataset includes a broad spectrum of error types, many of which are subtle (e.g., incorrect parameter values) and require semantic understanding of the task to identify. This reduces the risk that reward models succeed merely by detecting shallow patterns.
>     - Empirical validation: Figure 3 demonstrates a strong correlation (average r = 0.84) between reward model performance on FC-RewardBench and Best-of-N performance across five tool-use benchmarks, including real-user tasks such as API-Bank-1 and API-Bank-2. This strong correlation indicates that FC-RewardBench captures genuine task-relevant signal rather than superficial structural differences.
>     - These aspects of our methodology ensure and demonstrate that FC-RewardBench contains meaningful signals rather than superficial structural or pattern-based differences that have minimal impact on the overall outcome.
>
> - **For tool-calling reasoning, many scenarios can be constrained through rule-based outcome rewards**
>   - Rule-based methods, such as schema validation, can verify the syntactical properties of a tool call, such as tool and parameter hallucinations, type validity, inclusion of required parameters, etc. However, they cannot verify the semantic properties, such as incorrect parameter values, insufficient or unnecessary tool calls.
>   - In practice, models often fail due to semantic mistakes rather than simple syntactic errors, and rule-based checks are unable to capture these issues.
>   - In our FC-RewardBench experiments (**updated Figure 1**), the rule-based baseline (Schema-Validation Baseline) performs substantially worse than all learned reward models. We observe this same pattern across additional evaluations (**Figure 4**). This consistent performance gap highlights the limitations of rule-based methods for assessing nuanced tool-use behavior.
>
> - **Paper does not discuss process reward modeling approaches**
>   - We agree with the reviewer that process reward modeling (PRM) approaches for tool-calls are equally important.
>   - In **Section 2.3**, we already discuss both outcome and process reward modeling approaches along with the differences between them and our reasons to prefer outcome reward modeling over process reward modeling.
>
> - **Lack of direct application of the reward models to downstream tasks**
>   - Our original manuscript presents a data-filtering experiment where ToolRM is used to select the highest-quality 50% of training samples. Models fine-tuned on this filtered subset outperform those trained on the full corpus, demonstrating that ToolRM’s signal meaningfully improves downstream model quality.
>   - Following your suggestion (and reviewer 68SC’s), we have now **added a GRPO experiment in Section 5.3.2**. We train two models using three reward variants: (1) schema validation, (2) ToolRM rewards, and (3) ToolRL [1]’s correctness reward (requires gold outputs).
>     - We emphasize that both schema validation and ToolRM rewards require no ground truth, making them practical and realistic for RL settings, while ToolRL cannot be used when gold outputs are unavailable.
>     - Results on BFCL-v3 (**also available in the next comment**) show that ToolRM-based rewards match or exceed performance achieved using gold-truth-based rewards, further validating the effectiveness of ToolRM for RLHF and preference optimization.
>     - We thank the reviewer for this valuable suggestion, as these new results substantially strengthen the paper.
>
>
>
> We hope these revisions and clarifications fully address your concerns. We would be grateful for your reconsideration of the overall rating.
>
>
> [1] Qian, Cheng, et al. "Toolrl: Reward is all tool learning needs." arXiv preprint arXiv:2504.13958 (2025).

---

> ### Author Response · Authors · 2025-11-24
>
> Update:
>
> Dear Reviewer,
>
> We realized that we had missed a flag while running the evaluation script, which caused the reported numbers to be lower than they should have been. After re-running the evaluation with the correct flags, the results improved across all cases. Importantly, the conclusions from this experiment remain unchanged: *ToolRM-based rewards consistently match or exceed ToolRL-based (ground-truth–based) rewards, despite not requiring access to ground truth, highlighting their practicality and effectiveness as a scalable reward signal for reinforcement learning*. The updated table is included in the revised paper as well as below.
>
> We would be very happy to engage with you and answer any questions you may have about our work. We hope this clarification helps inform your assessment, and we sincerely appreciate your time and consideration. Looking forward to your reply.
>
> ----
>
> BFCL-v3 performance of Llama-3.2-3B-Instruct and Qwen2.5-3B-Instruct trained with GRPO under three reward designs
>
> |         Model         |  Reward Variant | Non-Live AST Acc | Live AST Acc |
> |:---------------------:|:---------------:|:----------------:|:------------:|
> | Llama-3.2-3B-Instruct |       Base      |      15.35%      |    43.82%    |
> |                       |     R_Schema    |      51.71%      |    62.25%    |
> |                       |     R_ToolRL    |      75.27%      |    64.25%    |
> |                       | R_ToolRM (Ours) |    **78.40%**    |  **64.32%**  |
> |  Qwen2.5-3B-Instruct  |       Base      |      43.06%      |    55.66%    |
> |                       |     R_Schema    |      63.17%      |    66.54%    |
> |                       |     R_ToolRL    |    **80.42%**    |    67.21%    |
> |                       | R_ToolRM (Ours) |      79.58%      |  **67.51%**  |

---

### Official Review · Reviewer_68SC · 2025-11-01

**Soundness:** 3
**Presentation:** 2
**Contribution:** 3
**Rating:** 4
**Confidence:** 5

**Summary:**

This paper introduces ToolRM, a family of outcome-based reward models specifically designed for evaluating and improving large language models in tool-calling scenarios. The authors also propose FC-RewardBench, the first benchmark for assessing reward models on function-calling correctness. ToolRM is trained on synthetic preference data derived from multiple open-weight function-calling models and evaluated across seven out-of-domain benchmarks. Experimental results show that ToolRM outperforms general-purpose reward models and LLM-as-a-Judge baselines.

**Strengths:**

1. The paper tackles an urgent and important problem: reward modeling for tool-calling LLMs.
2. The proposed FC-RewardBench is a well-constructed benchmark that fills a clear evaluation gap.
3. Empirical results show that ToolRM consistently outperforms general-purpose reward models on several tool-use benchmarks, and that it can be used effectively for Best-of-n inference and data filtering.

**Weaknesses:**

1. The motivation for designing a tool-specific, learned reward model is not clearly justified. Many aspects of tool correctness, such as format validity, AST equivalence, or parameter type checking, can already be handled by rule-based or symbolic evaluation metrics (which are very common). I suggest the author explain more limitations of these rule-based evaluations and further clarify why a learned LLM-based RM is necessary for this domain.

2. An implicit assumption is that a tool-specific reward model will lead to a more capable tool-use agent when used for RLHF or preference optimization. However, this is not demonstrated experimentally. It would greatly strengthen the paper to include a fine-tuning or RL experiment showing that ToolRM, as a reward signal, indeed improves tool-use reasoning beyond what general or rule-based rewards achieve.

3. The study focuses solely on tool-calling datasets. It remains unclear whether ToolRM generalizes beyond this domain. Evaluating on general-purpose reward modeling benchmarks such as RewardBench or RewardBench-2 would reveal whether the model’s learned signal captures general notions of quality or is over-specialized to tool-use syntax.

4. I am concerned that the experimental comparisons might not be fully fair. ToolRM is trained on datasets that include tool-use trajectories similar to those in the evaluation benchmarks, while the baseline reward models are not exposed to such data. I suggest the authors provide more clarification on the experiment setup to ensure a fair comparison.

**Questions:**

See above weakness

I recognize the authors' efforts in doing this work. However, there are some weaknesses that could be considered to strengthen the overall work.

---

> ### Author Response · Authors · 2025-11-24
> **Author response to Reviewer 68SC**
>
> We thank the reviewer for taking the time to review our work and provide valuable critical feedback. We are encouraged that you found the research problem important, FC-RewardBench well-constructed, and our empirical results strong. We address each of your concerns in detail below and have incorporated the corresponding improvements into the revised manuscript.
>
> - **Motivation for designing a tool-specific, learned reward model is not clearly justified**
>   - Rule-based methods, such as schema validation, can verify basic structural properties of a tool call such as tool and parameter hallucinations, type validity, inclusion of required parameters, etc. However, they miss certain key properties such as incorrect parameter values, insufficient or unnecessary tool calls.
>   - In practice, models often fail due to semantic mistakes rather than simple syntactic errors, and rule-based checks are usually unable to capture these issues.
>   - In our updated FC-RewardBench experiments (**updated Figure 1**), the rule-based baseline (Schema-Validation Baseline) performs substantially worse than all learned reward models. We observe this same pattern across additional evaluations (Figure 4).
>   - This consistent performance gap highlights the limitations of rule-based methods for assessing nuanced tool-use behavior.
>   - We have added these observations and clarifications to the revised manuscript to make the motivation clearer. Thank you for highlighting this to us.
>
> - **Fine-tuning or RL experiment**
>   - Our original manuscript presents a data-filtering experiment where ToolRM is used to select the highest-quality 50% of training samples. Models fine-tuned on this filtered subset outperform those trained on the full corpus, demonstrating that ToolRM’s signal meaningfully improves downstream model quality.
>   - Following your suggestion (and reviewer 3mfp’s), we have now **added a GRPO experiment in Section 5.3.2**. We train two models using three reward variants: (1) schema validation, (2) ToolRM rewards, and (3) ToolRL [1]’s correctness reward (requires gold outputs).
>     - We emphasize that both schema validation and ToolRM rewards require no ground truth, making them practical and realistic for RL settings, while ToolRL cannot be used when gold outputs are unavailable.
>     - Results on BFCL-v3 (**also available in next comment**) show that ToolRM-based rewards match or exceed performance achieved using gold-truth-based rewards, further validating the effectiveness of ToolRM for RLHF and preference optimization.
>     - We thank the reviewer for this valuable suggestion, as these new results substantially strengthen the paper.
>
> - **Unclear whether ToolRM generalizes beyond this domain**
>   - We would like to clarify that ToolRM is intentionally designed as a specialized reward model for function-calling, rather than as a general-purpose one.
>   - As shown in Figure 1, state-of-the-art general RMs fail significantly in the tool-based domain; therefore, we restricted our training strictly to tool-use data to bridge this specific gap.
>   - Consequently, we do not expect ToolRM to generalize to out-of-distribution tasks (like chat or safety in RewardBench), and this is by design. Instead, we focus solely on the tool-calling domain to:
>     - Highlight the failure of current RMs on tool-calling tasks;
>     - Propose an evaluation set (FC-RewardBench) that current benchmarks such as RewardBench miss,
>     - Propose a training framework specifically suited for the tool-calling domain; and
>     - Test the trained reward models on various benchmarks to show generalization and effectiveness
>
> - **Concerned that the experimental comparisons might not be fully fair**
>   - To the best of our knowledge, no existing reward models have been trained specifically for outcome-based evaluation of tool-calling behavior.
>   - ToolRM was introduced precisely to fill this gap, which required training on datasets containing tool-use trajectories.
>   - To further support the fairness of our evaluation, Figure 1 compares ToolRM not only against leading RewardBench RMs but also against state-of-the-art LLMs with strong tool-calling capabilities.
>   - Across these evaluations, ToolRM consistently outperforms both categories, demonstrating that a reward model specialized for tool calling delivers meaningful improvements beyond what existing general-purpose RMs or tool-capable LLMs can provide.
>
>
> We hope these revisions address all of your concerns. Your feedback has led to several meaningful additions - especially the RL experiments - that we believe strengthen the manuscript. We would be grateful for your reconsideration of the overall assessment in light of these substantial improvements.
>
> [1] Qian, Cheng, et al. "Toolrl: Reward is all tool learning needs." arXiv preprint arXiv:2504.13958 (2025).

---

> ### Author Response · Authors · 2025-11-24
>
> Update:
>
> Dear Reviewer,
>
> We realized that we had missed a flag while running the evaluation script, which caused the reported numbers to be lower than they should have been. After re-running the evaluation with the correct flags, the results improved across all cases. Importantly, the conclusions from this experiment remain unchanged: *ToolRM-based rewards consistently match or exceed ToolRL-based (ground-truth–based) rewards, despite not requiring access to ground truth, highlighting their practicality and effectiveness as a scalable reward signal for reinforcement learning*. The updated table is included in the revised paper as well as below.
>
> We would be very happy to engage with you and answer any questions you may have about our work. We hope this clarification helps inform your assessment, and we sincerely appreciate your time and consideration. Looking forward to your reply.
>
> ----
>
> BFCL-v3 performance of Llama-3.2-3B-Instruct and Qwen2.5-3B-Instruct trained with GRPO under three reward designs
>
> |         Model         |  Reward Variant | Non-Live AST Acc | Live AST Acc |
> |:---------------------:|:---------------:|:----------------:|:------------:|
> | Llama-3.2-3B-Instruct |       Base      |      15.35%      |    43.82%    |
> |                       |     R_Schema    |      51.71%      |    62.25%    |
> |                       |     R_ToolRL    |      75.27%      |    64.25%    |
> |                       | R_ToolRM (Ours) |    **78.40%**    |  **64.32%**  |
> |  Qwen2.5-3B-Instruct  |       Base      |      43.06%      |    55.66%    |
> |                       |     R_Schema    |      63.17%      |    66.54%    |
> |                       |     R_ToolRL    |    **80.42%**    |    67.21%    |
> |                       | R_ToolRM (Ours) |      79.58%      |  **67.51%**  |

---

### Official Review · Reviewer_sR8x · 2025-11-01

**Soundness:** 2
**Presentation:** 2
**Contribution:** 2
**Rating:** 2
**Confidence:** 5

**Summary:**

The paper proposes ToolRM, a family of outcome reward models (ORMs) tailored for the tool-calling regime, and introduces FC-RewardBench, a 1.5k-example benchmark designed to evaluate reward models’ ability to distinguish correct versus incorrect function calls. ToolRM models are trained using pairwise Bradley–Terry preferences formed from model-generated incorrect tool calls versus ground-truth calls. The paper claims (i) the first benchmark for reward models in function-calling, (ii) specialized ORMs that outperform domain-agnostic RMs and some LLM-judges at far smaller sizes, and (iii) strong correlation between FC-RewardBench accuracy and downstream tool-use performance.

**Strengths:**

1. The research problem is importtant. Rewarding validity of function calls is practically important for inference-time reranking and data filtering.

2. Outcome-only pairwise BT objective with reward centering and small backbones is easy to reproduce.

3. Introduce the FC-RewardBench evaluation with downstream tool-use performance targets a lower-cost proxy.

**Weaknesses:**

1. **Counter-evidence claims**: Tool-Augmented Reward Modeling (Themis, ICLR’24)  [1] explicitly equips RMs with external tool use and reports robust improvements, which aims to capture tool-based reasoning by letting the RM use tools during judgment. This directly contradicts the blanket claim that "existing RMs fail to capture" tool use the authors claimed in the introduction. The paper ignored previous tool-augmented RM literature.

2. **Factually incorrect novelty statement**. The statement “this is the first work introducing ORMs for tool calling” is demonstrably false.  This shows insufficient literature review on reward modeling for tool use.

3. **Missing comparison to tool-augmented RMs (IMPORTANT)**. The paper claims existing RMs fail to capture tool-based nuances, yet [1] have already studied tool-augmented reward models (TARM) across varying kinds of tools (covering both single-turn and multi-turn) and reports robust improvements on both RM and RL training. A fair assessment requires adding a TARM baseline (with tools enabled).

4. **Over-stated “no benchmark” claim**. The field has mature function-calling benchmarks (API-Bank, ToolBench/StableToolBench, BFCL) and increasingly mature RM benchmarks (RewardBench). If the novelty is “an RM-focused benchmark for pairwise outcome preferences in function calling,” this must be stated precisely.


## References:
[1] Tool-Augmented Reward Modeling. ICLR 2024.

**Questions:**

see above weakness.

---

> ### Author Response · Authors · 2025-11-24
> **Author response to Reviewer sR8x**
>
> We thank the reviewer for taking the time to review our work and provide valuable critical feedback. We are glad that you found the research problem important, the proposed benchmark a strength, and the technique easy to reproduce. We address each of your concerns in detail below and have incorporated the suggested improvements into the revised manuscript:
>
> - **Previous work and novelty statement**:
>   - Thank you for pointing us to the Tool-Augmented Reward Model (TARM) paper. We missed this important - though not counter-evidential - reference. We have added it to the revised paper along with a discussion of how it differs from our work.
>   - In particular, TARM and ToolRM serve fundamentally different purposes. TARM equips reward models to use tools to score an input, while ToolRM enables reward models to evaluate whether tool-call inputs are correct or not.
>   - As we show in the updated Figure 1, TARM struggles to reliably distinguish correct from incorrect tool-call outputs. Conversely, ToolRM is not designed to use tools to score inputs. We therefore view the two approaches as distinct, addressing complementary gaps in the literature.
>
> - **Missing comparison to tool-augmented RMs**
>   - Following your suggestion, we evaluated the Themis reward model (with tools enabled) on FC-RewardBench and have **added its results to Figure 1**. We note that Themis scores about 45% accuracy on the benchmark, significantly lower than other traditional reward models and roughly half the performance of ToolRM (90% on FC-RewardBench).
>   - This further supports our claim that while Themis effectively uses tools to score an input, it does not reliably discriminate between correct and incorrect tool-call inputs, reinforcing the need for ToolRM.
>
> - **Over-stated “no benchmark” claim: If the novelty is “an RM-focused benchmark for pairwise outcome preferences in function calling,” this must be stated precisely.**
>   - We explicitly and repeatedly state in the paper that FC-RewardBench is a benchmark for evaluating *reward models* in function-calling scenarios. For convenience, here are the exact citations:
>     - Abstract (Line 15): `FC-RewardBench, the first benchmark designed to systematically evaluate reward models in tool-calling scenarios`
>     - Introduction (Line 48): `no dedicated benchmark exists for evaluating reward models in the function-calling domain`
>     - Introduction (Line 79): `We introduce FC-RewardBench; the first benchmark to evaluate reward models on a tool calling setting`
>     - Methodology (Lines 145-147): `... there remains a notable gap in the evaluation of RMs for function-calling tasks. To bridge this gap, we propose FC-RewardBench, a benchmark specifically designed to evaluate RMs on function-calling tasks.`
>   - We also clearly acknowledge and distinguish between existing RM Benchmarks (like RewardBench) and FC-RewardBench. As stated in Methodology (Lines 144-145): `While several benchmarks evaluate RMs on tasks involving chat, reasoning, safety (Lambert et al., 2024); factuality, instruction following, and math (Malik et al., 2025), there remains a notable gap in the evaluation of RMs for function-calling tasks.`
>   - Furthermore, the function calling benchmarks you mentioned (BFCL-v3 and API-Bank), along with others, are already included as evaluation datasets in our experiments (see Table 2)
>   - Finally, we note that the novelty of this work is threefold: 1) FC-RewardBench, 2) a training framework for training tool calling reward models, and 3) a collection of reward models with varying sizes and their extensive evaluation against multiple baselines and in different inference settings.
>
> We hope these revisions and clarifications fully address your concerns. We believe the updated manuscript more clearly communicates the contributions and the importance of the problem setting. We would be grateful for your reconsideration of the overall rating.

---

> > ### Author Response · Authors · 2025-11-26
> >
> > Dear Reviewer,
> >
> > We would be very glad to engage in discussion, so please feel free to reach out with any questions or points that may benefit from further clarification. Thank you again for your time and thoughtful evaluation.

---

### Author Response · Authors · 2025-11-24
**Author response to all reviewers and area chair**

We thank all the reviewers for their time and effort in reviewing our paper. Your constructive feedback has led to several key improvements to the paper.

We are pleased to know that:
- Reviewers sR8x, 68SC, and p6AE found the research problem we tackle important
- Reviewers sR8x, 68SC, and 3mfp found the introduction of FC-RewardBench a significant contribution, and
- Reviewers 68SC and 3mfp found our experiments comprehensive and our empirical results strong.


We have addressed the specific concerns raised by the reviewers through clarifications and updates to the manuscript. Key points include:

- **Comparison to Tool-Augmented Reward Model (Reviewer sR8x):**
    - Reviewer sR8x asked for an empirical and conceptual comparison to Tool-Augmented Reward Models (TARM).
    - We clarified that TARM and ToolRM (ours) serve fundamentally different purposes addressing complementary gaps in the literature
    - We also **added TARM as a baseline in our FC-RewardBench experiment** (updated Figure 1). Results demonstrate that TARM struggles to reliably distinguish between correct and incorrect tool-call outputs.
- **Application in RL Training (Reviewers 68DC and 3mfp):**
    - Reviewers 68DC and 3mfp inquired about the application of ToolRM in RL training
    - We **added a new experiment (Section 5.3.2) training models using GRPO** with different reward variants. We show that models trained with ToolRM-based rewards match or exceed the performance of models trained with gold-truth-based rewards. This demonstrates the effectiveness and practicality of ToolRM as a scalable reward signal for reinforcement learning, without requiring access to ground truth.
- **Motivation for Learned Reward Models (Reviewers 68SC, 3mfp, and p6AE):**
    - Reviewers 68SC, 3mfp, and p6AE asked about the motivation to train a reward models instead of using a rule-based method
    - We clarified that rule-based methods can only validate syntactic tool-calling errors (e.g hallucinations, parameter type validity). They cannot, however, detect semantic errors, such as incorrect parameter value, or missing/unnecessary tool calls. This limitation necessitates a learned reward model.
    - We **updated Figure 1 to include a rule-based baseline (Schema Validation)**. The result show that this method is unable to distinguish between correct and incorrect tool calls and severely underperforms learned reward models.
- **Ablation study and ToolRM generalizability (Reviewer p6AE):**
    - We **added a new ablation study (Section 6)** in the paper, reporting model performance after ablating different hyperparameters, training dataset, and generator models
    - We **added a new section in Appendix A.6** showing low similarity between tools and conversations in the training and test set, indicating strong generalization to new tool types and conversations for ToolRM.

We believe we have adequately addressed the reviewers' concerns. We thank them again for their feedback, which has resulted in key improvements to the paper. Given these new experiments and clarifications, we hope the reviewers will reconsider their original ratings.

---

### Note · Authors · 2026-01-05

I have read and agree with the venue's withdrawal policy on behalf of myself and my co-authors.